# Evaluating glycerol dialkyl glycerol tetraether (GDGT)-based reconstructions from varved lake sediments during the Holocene

Ashley M. Abrook<sup>1,2</sup>, Gordon N. Inglis<sup>2</sup>, Peter G. Langdon<sup>1</sup>, McKenzie R. Bentley<sup>2</sup>, Achim Brauer<sup>3</sup>, Ian Bull<sup>4</sup>, Daisy Fallows<sup>1,2</sup>, Paul Lincoln<sup>5</sup>, Antti E. K. Ojala<sup>6,7</sup>, Helen L. Whelton<sup>4</sup>, Celia Martin-Puertas<sup>8</sup>

- <sup>1</sup> School of Geography and Environmental Science, University of Southampton, Southampton, UK
- <sup>2</sup> School of Ocean and Earth Science, University of Southampton, Southampton, UK
- <sup>3</sup> GFZ Helmholz Centre for Geosciences, Potsdam, Germany
- <sup>4</sup> School of Chemistry, University of Bristol, Bristol, UK
- <sup>5</sup> Department of Geography, Kings College London, London, UK
- <sup>6</sup> Department of Geography and Geology, University of Turku, Turku, Finland
- <sup>7</sup> Geological Survey of Finland, Espoo, Finland
- <sup>8</sup> Department of Geography, Royal Holloway University of London, Egham, UK
- 15 Correspondence to: Ashley M. Abrook (A.Abrook@soton.ac.uk)

Abstract. Advances in proxy development and proxy reconstructions within the Holocene increasingly reveal climatic complexity. Annually laminated (varved) lacustrine records provide an opportunity to assess this complexity at high temporal resolution. Organic geochemical proxies offer the potential for quantitative palaeoclimate reconstruction, however, their application to different varved lake settings remains limited. Here we explore the use of isoprenoid and branched glycerol dialkyl glycerol tetraethers (GDGTs) preserved in varved sediments as proxies for temperature. We analyse three Holocene-aged annually laminated lacustrine records spanning different climate regions and lake settings across Europe (Diss Mere, UK; Nautajärvi, Finland; Meerfelder Maar, Germany); including intervals at multi-decadal resolution within the mid-Holocene. We show that isoprenoid GDGT distributions in annually laminated sediment sequences are largely derived from methanogenic Euryarchaeota and yield unreliable lake surface temperature reconstructions. Conversely, branched GDGT reconstructions show good coherence with instrumental temperature data in mid- and high-latitude environments. Although we show that lake or catchment-specific processes, including differences in processes linked to varve sedimentation, hypoxia, sediment influx and landscape development, can influence brGDGT distributions in varved lakes, the trends and range of variability of our brGDGT derived Holocene temperature reconstructions broadly agree with regional European Holocene reconstructions. This suggests that temperature exerts a first-order control on the methylation of brGDGTs in varved lake sequences. Combined with precise varve chronologies, these biomarker records can be used to generate highly resolved climate data across the Holocene.

Keywords: isoprenoid and branched GDGTs, biomarkers, temperature reconstructions, varves, Holocene

#### 1 Introduction

The Holocene (11.7 ka BP to present) has previously been considered as climatically stable when viewed alongside the Glacial-Interglacial cycles of the Quaternary. However, with increasing temporal and spatial resolution of proxy data, the Holocene reveals: a) rapid (multi-decadal) shifts in climate systems associated with abrupt climate events (e.g., Roland et al., 2015; Parker and Harrison, 2022; Candy et al., 2025), b) a disconnect between modelled climate evolution and proxy-based climate reconstructions (Liu et al., 2014; 2025; Kaufman et al., 2020; Osman et al., 2021), and c) spatial changes to 40 dominant modes of climatic variability through time (e.g., the North Atlantic Oscillation; Hernández et al., 2020). To disentangle this climatic complexity, highly resolved climate archives are required. Whilst internationally important archives with excellent temporal resolution exist, frequently these are limited in terrestrial palaeo- studies due to temporal limitations (e.g., tree rings; Briffa et al., 2004) and/or restricted in their geographic location (e.g., ice-cores; Rasmussen et al., 2014, and speleothems; Wang et al., 2005). A complementary archive that offers the potential to disentangle climatic complexity across different regions over long timescales are annually laminated (varved) lake sediments, with an absence of sedimentary mixing. These sediment sequences contain precise chronologies obtained via annual layer counting tied to radiocarbon and/or tephra frameworks (e.g., Lane et al., 2013; Zolitschka et al., 2015, Wulf et al., 2015). The analysis of climatic proxies within these sediments (e.g., Ojala et al., 2008; Larocque-Tobler et al., 2015; Luoto and Ojala et al., 2016; Lincoln et al., 2025) offers the potential for highly resolved (annual to multi-decadal) past climate reconstructions across wide spatial scales. Here, biomarkers are of particular interest as they can provide quantified climatic and environmental observations from lakes (Rach et al., 2014) yet the degree to which biomarkers faithfully record past temperature in varved lake sediment archives remains an open question.

Isoprenoid and branched glycerol dialkyl glycerol tetraethers (GDGTs) are membrane-spanning lipids and are frequently used to reconstruct climate variability in lake sediments (e.g., Tierney and Russell, 2009; Woltering et al., 2014; Peaple et al., 2022; Robles et al., 2023; Baxter et al., 2024; Zander et al., 2024). Isoprenoid GDGTs are archaeal lipids composed of two isoprenoid alkyl chains with up to four cyclopentane moieties identified in lakes (Schouten et al., 2013). The degree of isoGDGT cyclisation is captured *via* the TetraEther indeX of 86 carbons (TEX<sub>86</sub>) which is calibrated to surface water temperature in lacustrine settings (Powers et al., 2010). Branched GDGTs are bacterial lipids (Sinninghe Damsté et al., 2000) composed of two branched alkyl chains, varying in the number of methyl branches (4 to 6) and cyclopentane moieties (0 to 2) (Weijers et al., 2007). The degree of methylation is captured *via* the methylation index of branched tetraethers (MBT'<sub>SME</sub>; De Jonge et al., 2014), which in lakes is calibrated to either mean annual air temperature (MAAT; Russell et al., 2018), mean temperature of months above freezing (MAF; Martinez-Sosa et al., 2021; Raberg et al., 2021) and/or mean summer temperatures (MST; Pearson et al., 2011; 2025).

75

Whilst GDGTs have been applied extensively to Holocene-aged lake sediments, few studies have evaluated the environmental controls on GDGT distributions within annually laminated lake sediments in Europe (i.e., Weber et al., 2018; Zander et al., 2024; Otiniano et al., 2024). These settings exhibit specific limnological processes, including hypolimnetic anoxia/hypoxia, thermal stratification and individual mixing regimes that may exert strong controls upon the distribution of different GDGT compounds. Some of these features (e.g., seasonal anoxia), alongside lake morphometry and catchment size, are known to affect the ecology of archaeal and bacterial groups that synthesise isoprenoidal and branched GDGTs, respectively (Weber et al., 2018; van Bree et al., 2020; Baxter et al., 2024; Zander et al., 2024). As these proxies are becoming more common in varve-based palaeoclimate research, an evaluation is required to establish whether annually laminated lake records from different climatic settings and/or the processes that govern lamination sedimentology (e.g., clastic, organic, diatomaceous, carbonate lamina) impact GDGT distributions and associated metrics.

Here, we use three different annually laminated lake records from Europe to establish whether GDGT distributions in varved lake sediments are suitable for generating climate reconstructions during the Holocene. We assess whether processes that 1) are important for varve formation and preservation (e.g., stratification and anoxia) and 2) control varve composition, impact GDGT distributions and metrics. Isoprenoid and branched GDGTs are analysed at multi-decadal resolutions across key intervals of the Holocene, including the Holocene Thermal Maximum (ca. 6.5 – 5 ka BP) and the last 300 years. We compare our results with instrumental data and other independent proxies to provide a unique assessment of GDGT distributions in varved lake systems and evaluate the potential of using isoGDGTs and brGDGTs to reconstruct highly resolved temperature records across the Holocene and beyond.

#### 85 2 Materials and Methods

## 2.1 Site descriptions

#### 2.1.1 Diss Mere

90

Diss Mere (52° 22′N, 1° 6′E; Fig. 1) is a small (0.034 km²), 6 m deep, presently eutrophic lake with alkaline waters and a chalk bedrock (Martin-Puertas et al., 2021) which occupies a region dominated by maritime climate. During lake stratification, pH is high in the epilimnion (values of ~9) and lower in the hypolimnion (~7) (Boyall et al., 2023). The lake contains no inflow or outflow, experiences minimal groundwater input (Boyall et al., 2023) and has an estimated catchment to lake area ratio (CA:LA) of 44. The DISS-16 record exhibits 15 m of sedimentation covering the last 10 thousand years (Martin-Puertas et al., 2021) with varves preserved between 10 and 2 ka BP. The varve couplets are calcite-organic (Fig. 1; including diatoms in the organic component) with some interannual variability (Martin-Puertas et al., 2021). The calcite layer is representative of summer sedimentation with organic constituents deposited during autumn to spring following

resuspension of organic remains and wind-induced mixing of the water column (Martin-Puertas et al., 2021; Boyall et al., 2023).

#### 2.2.2 Nautajärvi

100

105

Nautajärvi (61°48'N 24°24'E; Fig. 1) is a small (0.17 km²), ~20 m deep mesotrophic lake, exhibiting slightly acidic lake waters (modern pH of 5.8 – 6; Korkonen et al., 2017), which occupies a region of sub-Arctic climate. The site sits in a tiered lake system within the Äväntäjärvi drainage basin and is fed by streams and Ristijärvi to the north and is drained by Lake Pitkävesi to the south (Ojala and Alenius, 2005) with a CA:LA of 71. The NAU-23 stratigraphy (Lincoln et al., 2025) contains 7.3 m of sedimentation, with 9,829 varves covering the upper 6.74 m (Ojala and Saarinen, 2002; Lincoln et al., 2025). The varve couplets are clastic-organic (Fig. 1), with the clastic layer forming in spring as a product of winter precipitation and snowmelt run off once the lake becomes ice-free, with the organic layer forming in the summer to winter months (Ojala and Alenius, 2005). Inverse stratification is observed during the winter months with anoxic bottom waters at this time (Ojala and Alenius, 2005; Ojala et al., 2013).

#### 2.2.3 Meerfelder Maar

Meerfelder Maar (50°06′N, 6°45′E; Fig. 1) is a relatively small (0.25 km²), 18 m deep, eutrophic maar lake which occupies a predominantly maritime climate region (albeit with some continental influences) within mainland Europe. The lake has a CA:LA of 6. The 11.71 m long MFM-09 sequence contains 7.45 m of sediment across the Holocene (Martin-Puertas et al., 2012). Varves are preserved from 11.6 to ca. 1.5 ka BP (Martin-Puertas et al., 2012; 2017). The Holocene varve structures preserved at Meerfelder Maar are diatomaceous (Fig. 1), with a spring and summer layer composed of a monospecific diatom bloom and the autumn and winter layer composed of organic and minerogenic detritus (Brauer et al., 2000; Martin-Puertas et al., 2012). Hypolimnetic anoxia has previously been proposed at Meerfelder Maar, with treatments performed to increase bottom water oxygen conditions (Nürnberg et al., 2007). This process disturbed stratigraphy of the upper 500 years of the record.

Table 1: Lake descriptors from each site. \* This is the assumed mixing regime for periods of varve formation across each site, at 120 Diss Mere and Meerfelder Maar this is likely different to today. \*\* The origin of Diss Mere is debated, with possible origins dating back to the Anglian stage (Bailey, 2005).

| Lake            | Climatology    | Lake origin    | Mixing regime* | Varve composition |
|-----------------|----------------|----------------|----------------|-------------------|
| Diss Mere       | Maritime       | Thermokarst**  | Meromictic     | Calcite – organic |
| Nautajärvi      | Sub-Arctic     | Glacial lake   | Dimictic       | Clastic – organic |
| Meerfelder Maar | Maritime (with | Volcanic (maar | Meromictic     | Diatomaceous –    |
|                 | continental    | lake)          |                | detrital          |
|                 | influences)    |                |                |                   |

## 2.2.4 Chronology

Published chronologies are derived from annual layer counting and tied to either a radiocarbon and/or tephra framework (Martin-Puertas et al., 2012; 2021; Meerfelder Maar, Diss Mere respectively) or from the cross-correlation of marker layers between an annual layer-counted sequence and newly derived sediments (e.g., Ojala and Saarinen, 2002; Ojala and Alenius, 2005; Lincoln et al., 2025; Nautajärvi). For each sequence the chronological uncertainties are decadal in scale (see Matin-Puertas et al., 2012; 2021; Lincoln et al., 2025). Following Abrook et al. (2025), we present reconstructions as ka BP (as cal. ka BP at Diss Mere and Meerfelder Maar and ka varve years BP for Nautajärvi).

Figure 1: Location of each of the sites in Europe: a) Diss Mere, UK, b) Nautajärvi, Finland; c) Meerfelder Maar, Germany. The bottom panel shows the dominant laminations from each of the site which together contribute to 1-year of sedimentation (Diss Mere adapted from Boyall et al., 2023; Nautajärvi adapted from Lincoln et al., 2025).

#### 135 2.2 Biomarker analysis

Biomarker samples, 0.5 cm resolution at variable intervals, were extracted from the across the Holocene and from modern sediments (covering the last ca. 300 years) from Diss Mere (n = 56) and Nautajärvi (n = 42). Due to historic hypolimnetic treatments at Meerfelder Maar (n = 31) the last 500 years of sedimentation have been impacted. Analyses were therefore not performed on modern sediments at this site. To obtain total lipid extracts (TLE), approximately 1 - 2g of wet sediment per sample was freeze dried and extracted using Accelerated Solvent Extraction (Dionex ASE 350) with dichloromethane:methanol (9:1, v/v) at  $100^{\circ}$ C over 3 x 15 min extraction cycles at 1,500 PSI. The TLE was dried under a stream of nitrogen (N<sub>2</sub>) and separated into different compound classes via silica gel column chromatography (F1; aliphatic, F2; aromatic, and F3; polar) using hexane:dichloromethane (9:1, v/v; F1), hexane:dichloromethane (1:1, v/v; F2), and dichloromethane:methanol (1:1, v/v; F3) with a 4ml solvent volume for each fraction. The polar fraction was re-dissolved in hexane:isopropanol (99:1, v/v) and passed through a 0.45 µm PTFE (polytetrafluoroethylene) filter and analysed by high-performance liquid chromatography (HPLC) mass spectrometry (MS) using an Agilent 1260 HPLC coupled to an Agilent 6130 single quadrupole MSD following the method outlined in Hopmans et al. (2016) (supplement). Peak identification and GDGT integration followed a manual process with identification aided using in-house GDGT laboratory standards after scanning for common mass ranges for GDGT analysis (even values between m/z 1302 – 1292 for isoGDGTs and between (1022 – 1018, 1036 – 1032 and 1050 – 1046 for brGDGTs).

# 2.3 Biomarker analysis

#### 2.3.1 GDGT-based temperature proxies

 $TEX_{86}$  is defined following Schouten et al. (2002).

$$TEX_{86} = \frac{(GDGT - 2 + GDGT - 3 + Crenarchaeol regioisomer)}{(GDGT - 1 + GDGT - 2 + GDGT - 3 + Crenarchaeol regioisomer)}$$
(1)

TEX<sub>86</sub> is calibrated to Lake Surface Temperature (LST) following Powers et al. (2010). See supplement.

$$LST = 55.2 \cdot TEX_{86} - 14$$
 (2)

We use the MBT'<sub>5ME</sub> index following De Jonge et al. (2014).

$$MBT'_{5ME} = \frac{(Ia + Ib + Ic)}{(Ia + Ib + Ic + IIa + IIb + IIc + IIIa)}$$
(3)

Where alphanumeric values relate to different brGDGT structures and the number of cyclopentane moieties and/or methyl branches. MBT'<sub>5ME</sub> values are converted to mean temperature of months above freezing (MAF; Martinez-Sosa et al., 2021) using a Bayesian calibration (BayMBT<sub>0</sub>) developed for lakes (e.g., Martinez-Sosa et al., 2021). Present day mean annual temperatures are used as prior means at each site: 10.5°C (Diss Mere); 5.3°C (Nautajärvi) and 9.9°C (Meerfelder Maar). Prior standard deviation is set to 10°C.

We also use the multivariate calibration approach of Raberg et al. (2021).

$$MAF = 92.9 (\pm 15.98) + 63.84 (\pm 15.58) \cdot \text{flb}_{\text{meth}}^2 - 130.51 (\pm 30.73) \cdot \text{flb}_{\text{meth}} - 28.77 (\pm 5.44) \cdot \text{flIa}_{\text{meth}}^2 - 72.28 (\pm 17.38) \cdot \text{flIb}_{\text{meth}}^2 - 5.88 (\pm 1.36) \cdot \text{flIc}_{\text{meth}}^2 + 20.89 (\pm 7.69) \cdot \text{flIIa}_{\text{meth}}^2 - 40.54 (\pm 5.89) \cdot \text{flIIa}_{\text{meth}} - 80.47 (\pm 19.19) \cdot \text{flIIb}_{\text{meth}}$$

$$(4)$$

#### 2.3.2 GDGT-based lake property reconstructions

The cyclisation of branched tetraethers (CBT) index (Weijers et al., 2007) that includes both 5- and 6-methyl isomers (De Jonge et al., 2014) is used to infer lake pH:

$$CBT' = {}^{10}\log\left(\frac{(\operatorname{Ic} + \operatorname{IIa'} + \operatorname{IIb'} + \operatorname{IIc'} + \operatorname{IIIa'})}{(\operatorname{Ia} + \operatorname{IIa} + \operatorname{IIIa})}\right)$$
(5)

We subsequently convert CBT' values to pH using the calibration from Russell et al. (2018).

$$pH = 8.95 + 2.65 \cdot CBT'$$
 (6)

The isomer ratio captures the relative abundance of 5-methyl vs 6-methyl brGDGTs (De Jonge et al., 2015) and is defined as:

$$IR_{6ME} = \frac{(IIa' + IIb' + IIc' + IIIa' + IIIb' + IIIc')}{(IIa + IIa' + IIb' + IIc + IIc' + IIIa' + IIIb' + IIIc' + IIIc')}$$

$$(7)$$

Deviations from the expected temperature –  $MBT_{5ME}$  relationship are observed in lake sediments when the  $IR_{6ME} > 0.5$  (Bauersachs et al., 2024).

We use the methane index (MI) to evaluate anaerobic methanotrophy within the lake water column and/or sediments, whereby sediments with high values (>0.3 - 0.5) suggest intense anaerobic methane oxidation (Zhang et al., 2011).

$$MI = \frac{(GDGT-1 + GDGT-2 + GDGT-3)}{(GDGT-1 + GDGT-2 + GDGT-3 + Crenarchaeol + Crenarchaeol regioisomer)}$$
(8)

Crenarchaeol and its regioisomer are solely produced by *Nitrososphaeria* (Zhang et al., 2011; Schouten et al., 2013), whereas GDGT-0 can also be produced by methanogenic archaea (Euryarchaeota) (Auget et al., 2009). The relative abundance of GDGT-0 vs Crenarchaeol (i.e. % GDGT-0) is thus used to evaluate methanogenesis across each of the different lakes.

$$\% GDGT - 0 = \left( \left( \frac{GDGT - 0}{GDGT - 0} \right) + Crenarchaeol \right) \cdot 100$$
(9)

Where values > 67% are considered to have a substantial methanogen component that could render TEX<sub>86</sub> reconstructions unreliable (e.g., Blaga et al., 2009; Inglis et al., 2015).

# 190 2.3.3 GDGT-based organic source proxies

The branched vs isoprenoid tetraether (BIT) index evaluates the relative proportion of terrestrial vs marine organic matter (OM) within individual samples and is defined as:

$$BIT = \frac{(\text{Ia} + \text{IIa} + \text{IIIa} + \text{IIIa} + \text{IIIa})}{(\text{Ia} + \text{IIa} + \text{IIIa} + \text{IIIa} + \text{IIIa} + \text{Crenarchaeol})}$$
(10)

Where high values (e.g., 1) traditionally reflect high terrestrial input and low values (e.g., 0) indicate high marine/aquatic input.

The ratio of hexamethylated and pentamethylated compounds (Xiao et al., 2016) is used to assess input of terrestrial vs marine-derived brGDGTs, where  $\Sigma$ IIIa /  $\Sigma$ IIa values <0.59 suggest terrestrial provenance and high values >0.92 represents a marine/aquatic provenance (Xiao et al., 2016).

Finally, we use the branched and isoprenoid GDGT machine learning classification algorithm (BIGMaC) to constrain GDGT source environments (Martinez-Sosa et al., 2023). The algorithm assesses the distribution of all isoGDGTs and brGDGTs and establishes (dis)similarity to GDGT distributions from a global marine, soil, lake, and peat calibration dataset.

#### 2.4 Micro X-ray Fluorescence (μ-XRF) core scanning data

To assist in disentangling environmental processes, geochemical element profiles of each sequence (DISS-16, NAU-23, MFM-09) have been acquired directly from split-core surfaces every 0.2 mm using an ITRAX μ-XRF core scanner at GFZ-Potsdam. Element data has already been published (Martin-Puertas et al., 2012; Boyall et al., 2024; Lincoln et al., 2025), however, to aid interpretations we re-sample those data to match the ages of the GDGT data.

## 2.5 Data analyses

Ternary diagrams were created in R through proportional fractional abundances of tetra-, penta- and hexamethylated compounds using the 'ggtern' package (Hamilton and Ferry, 2018). Unconstrained and constrained ordination (principal components analysis; PCA, and redundancy analysis; RDA) was performed on standardised brGDGT fractional abundance data. We used PCA and RDA owing to short gradient lengths observed within the GDGT data, which were determined in R. For RDA, explanatory variables consisted of the major elements from centered log ratio μ-XRF data (e.g., Fe<sub>clr</sub>) from each lake. PCA and RDA analysis was performed in R using the 'factoextra' and 'Vegan' packages (Kassambara, 2023; Oksanen et al., 2024). Correlation plots are shown in supplement.

#### 3 Results

#### 3.1 GDGT distributions and GDGT-based environmental metrics

isoGDGTs were identified in all samples from Diss Mere, Meerfelder Maar and Nautajärvi. GDGT-0 dominates the isoGDGT assemblage at all three site locations, across modern and Holocene samples (mostly > 90% at Diss Mere and Meerfelder Maar, > 80% at Nautajärvi), with a lower relative abundance of GDGT-1 to -3, Crenarchaeol and its regioisomer (Fig. 2). Each site exhibits high MI values with values averaging  $0.73 \pm 0.21$ ,  $0.70 \pm 0.10$  and  $0.64 \pm 0.09$ , respectively.

brGDGTs were also identified in all samples. However, there are differences in the relative proportion of tetra-, penta- and hexamethylated brGDGTs between sites (Fig. 3). Nautajärvi and Meerfelder Maar have a higher contribution of 5-methyl brGDGTs than Diss Mere, with a higher relative proportion of penta- and hexamethylated compounds at Nautajärvi (e.g., brGDGT-IIa, IIIa). Diss Mere and Meerfelder Maar have a larger relative proportion of 6-methyl brGDGTs (e.g., brGDGT-IIa', IIIa') than Nautajärvi. This is reflected in low average IR<sub>6ME</sub> values at Nautajärvi (0.15 ± 0.04), with higher values at Meerfelder Maar (0.41 ± 0.03) and Diss Mere (0.60 ± 0.06). The ΣIIIa / ΣIIa ratio averages 0.74 ± 0.07 at Diss Mere, 0.64 ± 0.07 at Nautajärvi, and 0.55 ± 0.08 at Meerfelder Maar.

The BIGMaC algorithm classified 100% of GDGT distributions from Diss Mere and Meerfelder Maar and 67% of distributions from Nautajärvi as 'lake-type'. The remaining 33% of GDGTs from Nautajärvi (i.e., all modern [< 300 year]

and two late-Holocene samples) are classified as 'peat-type'. Across all three sites the BIT index is very high and is > 0.92.

## 235 3.2 Statistical analyses

PC1 and PC2 explain 39.3% and 25.7% of variation at Diss Mere, 56.1% and 23.9% of variation at Nautajärvi, and 38.8% and 23.2% of variation at Meerfelder Maar, respectively (Fig. 4). PCA shows that 5- and 6-methyl brGDGTs are separated at Diss Mere but not at Nautajärvi or Meerfelder Maar. At the latter two sites, however, the brGDGTs with the largest fractional abundance (i.e., brGDGT-Ia, IIa) are separated from the remainder of brGDGTs. PCA reveals that at Diss Mere the modern and Holocene samples occupy a similar ordination space whilst at Nautajärvi the modern samples are adjunct from the remainder of the data.

RDA1 and RDA2 explain 52.2% and 32.6% of variation at Diss Mere, 84.1% and 9.1% of variation at Nautajärvi, and 51.9% and 20.8% of variation at Meerfelder Maar, respectively. At each site, redox sensitive elements appear important (e.g., Fe<sub>clr</sub>, Mn<sub>clr</sub>, S<sub>clr</sub>) with additional importance given to the relationship between authigenic (e.g., Ca<sub>clr</sub>) and allogenic components (K<sub>clr</sub> and Ti<sub>clr</sub>) between sites.

Figure 2: Fractional abundances of different GDGT molecules in the different lakes. a) isoprenoid GDGT distributions; b) branched GDGT distributions.

Figure 3: Ternary plots showing the relative proportions of tetramethylated, pentamethylated and hexamethylated compounds from Diss Mere, Meerfelder Maar and Nautajärvi, respectively against the same data from a global brGDGT peat and soil dataset (grey; Dearing Crampton-Flood et al., 2020); a global lake dataset (black; Martinez-Sosa et al., 2021); Arctic lakes (green; Raberg et al., 2021) and Alpine lakes (red; Bauersachs et al., 2024).

#### 3.3 Temperature and pH reconstructions

TEX<sub>86</sub> has a larger range of values at Diss Mere (average of 0.41 ± 0.13) compared to Meerfelder Maar (0.30 ± 0.04) and Nautajärvi (0.37 ± 0.08) (Figs. 5, 6, 7). When converted to LST the data from each site appears variable, with greatest range in LST observed at Diss Mere (-9.0°C to 25°C), Nautajärvi (1°C to 14°C) and Meerfelder Maar (-1°C to 7°C) (S1).

The lowest average MBT'<sub>5ME</sub> values are from Nautajärvi ( $0.36 \pm 0.02$ ) whilst Diss Mere ( $0.44 \pm 0.04$ ) and Meerfelder Maar ( $0.45 \pm 0.02$ ) exhibit higher values (Figs. 5, 6, 7). brGDGT temperatures using the Bayesian (Martinez-Sosa et al., 2021) and multivariate calibration (Raberg et al., 2021) approaches produce mostly similar trends across each site. Lowest absolute temperatures are generated from Nautajärvi with temperatures between 8°C and 12°C across all samples. Reconstructed temperatures from Diss Mere range from 9°C to 15°C. At Meerfelder Maar temperature estimates range between 11 to 14°C.

The CBT' index is highest at Diss Mere (0.05 ± 0.07) and lowest at Nautajärvi (-0.94 ± 0.15). This yields higher pH reconstructions at Diss Mere (9.08 ± 0.19) and lower pH estimates at Nautajärvi. (6.51 ± 0.39). Meerfelder Maar exhibits an intermediary pH reconstruction (8.11 ± 0.15) (Figs. 5, 6, 7).

Figure 4: Principal components analysis (PCA) left, and redundancy analysis (RDA) right. Panel a) details the plots from Diss Mere, panel b) from Nautajärvi and panel c) from Meerfelder Maar. Panels a) and b) show modern and Holocene distributions within the PCA plots. Explanatory variables in the RDA plots are major elements from  $\mu$ -XRF data (e.g., Martin-Puertas et al., 2012; Boyal et al., 2024; Lincoln et al., 2025).

#### **4.0 Discussion**

#### 4.1 isoGDGTs influenced by methanogenesis in varved lake systems

In lacustrine environments, the TetraEther indeX of 86 carbons (TEX<sub>86</sub>) has previously been used to infer Lake Surface Temperature (LST; Woltering et al., 2012). This assumes that isoGDGTs are mostly derived from ammonia-oxidizing archaea of the class *Nitrososphaeria*. However, methanogenic archaea may also synthesise smaller quantities of isoGDGTs with up to three cyclopentane moieties (Weijers et al., 2006), although this has not been confirmed in culture studies (Schouten et al., 2013). This implies that input of methanogenic archaea could bias lacustrine TEX<sub>86</sub> values (Blaga et al., 2009), especially in varved lakes where seasonal anoxia/hypoxia is a persistent feature. Here, the fractional abundance of GDGT-0 is elevated at all sites (Diss Mere > 0.84 [mostly >0.9], Nautajärvi > 0.81 [excluding one sample of 0.62], and Meerfelder Maar > 0.94) and associated with very high % GDGT-0 values (>67 %; see Inglis et al., 2015). These values suggest that isoGDGTs are not produced by ammonia-oxidizing archaea but instead synthesized mostly by methanogens.

Previous studies show that lakes that exhibit hypolimnetic anoxia have a higher abundance of GDGT-0 in sediments compared to the water column, implying that GDGT-0 is largely derived from methanogens present within the sediment (Blaga et al., 2009). However, in large lakes, Baxter et al. (2021) and Sinninghe Damsté et al. (2012) demonstrate that GDGT-0 can also be synthesised in high concentrations below the oxycline, whereas *Nitrososphaeria* synthesized abundant Crenarchaeol in relatively oxygenated upper waters. No sediment trap or water column data is available for this manuscript, but, as each lake exhibits hypolimnetic anoxia/hypoxia (e.g., Brauer et al., 2000; Ojala and Alenius, 2005; Martin-Puertas et al., 2021; Boyall et al., 2023), it is likely that isoGDGTs are largely produced by methanogenic Euryarchaeota. Given the inconsistent variability in LST between sites (Section 3.3), and, in some instances, an extreme range for Holocene climates (e.g., at Diss Mere, with reconstructions from  $-9.0^{\circ}$ C to  $25^{\circ}$ C; S1), this suggests that a methanogen overprint is highly likely.

Anaerobic methanotrophs (ANME) can also synthesise GDGT-0 to GDGT-3 and may contribute to the observed high Methane Index (MI) values (Zhang et al., 2011; Kim and Zhang, 2023). These organisms are typically abundant in sites characterized by anaerobic oxidation of methane (AOM, e.g., cold seeps, mud volcanoes, gas hydrates) and require high concentrations of sulphate in the overlying water column (Zhao et al., 2024). Although sulphate is often variable in lakes, sulphate reduction can be maintained in freshwater systems (Segarra et al. 2015). At each location in our study, μ-XRF data reveals varying contributions of elemental sulphur (with relationships with GDGTs; S3 – S8) which, given hypoxic conditions, can lead to sulphate reduction (albeit strongly reducing conditions may not be maintained throughout the annual cycle; Lincoln et al., 2025). These features, alongside high MI values, may suggest AOM-impacted sediments. High MI and % GDGT-0 values are not surprising as the two proxies are linearly correlated in marine sediments (Inglis et al., 2015) and

Figure 5: Main GDGT reconstructions from Diss Mere across the Holocene. Shown are a) TEX86; b) MBT'5ME (De Jonge et al., 2014); c) CBT' (De Jonge et al., 2014); d) reconstructed pH (Russell et al., 2018); e)  $\Sigma$ IIIa /  $\Sigma$ IIIa ratios with cut offs from Xiao et al. (2016); f) IR6ME (De Jonge et al., 2014; Bauersachs et al., 2024); g) the methane index (Zhang et al., 2011); h) % GDGT-0 and i) the branched vs isoprenoid tetraether index (BIT).

Figure 6: Main GDGT reconstructions from Nautajärvi across the Holocene. Shown are a)  $TEX_{86}$ ; b) MBT'<sub>5ME</sub> (De Jonge et al., 2014); c) CBT' (De Jonge et al., 2014); d) reconstructed pH (Russell et al., 2018); e)  $\Sigma$ IIIa /  $\Sigma$ IIa ratios with a cut off from Xiao et al. (2016); f) IR<sub>6ME</sub> (De Jonge et al., 2014); g) the methane index (Zhang et al., 2011); h) % GDGT-0 and i) the branched vs isoprenoid tetraether index (BIT).

Figure 7: Main GDGT reconstructions from Meerfelder Maar across the Holocene. Shown are a) TEX86; b) MBT'5ME (de Jonge et al., 2014); c) CBT' (de Jonge et al., 2014); d) reconstructed pH (Russell et al., 2018); e)  $\Sigma$ IIIa /  $\Sigma$ IIa ratios with a cut off from Xiao et al. (2016); f) IR6ME (De Jonge et al., 2014; Bauersachs et al., 2024); g) the methane index (Zhang et al., 2011); h) % GDGT-0 and i) the branched vs isoprenoid tetraether index (BIT).

lakes (Collins et al., 2025). However, the relative contribution of methanogens and methanotrophs to the archaeal community is unknown and would require further isotopic analysis (e.g., Segarra et al., 2015). Overall, it is apparent that methane cycling can bias isoGDGT distributions in lakes (this study; Blaga et al., 2009; Weijers et al., 2011; Zhang et al., 2016; Baxter et al., 2021) and we caution against the use of TEX<sub>86</sub> in varved lake systems.

# 4.2 brGDGTs likely derived from in-situ lacustrine production in varved lakes

brGDGTs in lakes can have multiple sources including input *via* soils/peats (e.g., Peterse et al., 2014) and *in situ* production within the lake water column (e.g., Buckles et al., 2014). It is therefore important to understand the provenance of brGDGTs in lacustrine environments as different soil, peat, and lake calibrations (e.g., Russell et al., 2018; Dearing Crampton-Flood et al., 2020; Martinez-Sosa et al., 2021; Raberg et al., 2021) can yield temperature estimates that can differ by up to 10°C (Tierney et al., 2010). This is likely to be important in annually laminated lakes where varve sediment composition may be reflective of changes in lacustrine or catchment productivity and changes in autochtonous process (e.g., primary productivity in the water column and suspension settling of mineral or organic particles vs in wash).

Branched *vs* isoprenoid tetraether (BIT) values are historically interpreted to reflect changes in delivery of soil/peat derived organic matter into aquatic systems (e.g., Blaga et al., 2009; Sinninghe Damsté et al., 2009; Schouten et al., 2013). Across all three lakes, the BIT index is high for all samples (> 0.9), which implies consistently high input of soil or peat organic matter into the lake. Each lake is relatively small (< 0.25 km²), and depositional models for detrital and clastic lamination types indicate allochthonous input *via* catchment runoff limited to specific seasons (Meerfelder Maar, Brauer et al., 2000; Nautajärvi, Lincoln et al., 2025). Allied to this, each lake has different catchment to lake area (CA:LA) ratios which might impact BIT if viewed solely from a catchment provenance perspective. However, increasingly evidence suggests that higher BIT values can result from relatively low Crenarachaeol concentrations in lakes (Loomis et al., 2011; Buckles et al., 2014; van Bree et al., 2020) and/or *in-situ* production of brGDGTs (Naeher et al., 2014; Buckles et al., 2014). Therefore, these observations suggest caution when using the BIT to evaluate terrestrial *vs* aquatic input in annually laminated lake sediments.

differentiate between soil vs aquatic input. This approach has mostly been applied in marine sediments (Xiao et al., 2016), but Martin et al. (2019) suggested that  $\Sigma$ IIIa / $\Sigma$ IIa ratios can also distinguish between soil vs aquatic input in lacustrine settings. All Holocene samples from Diss Mere and Nautajärvi (except one sample from Nautajärvi at 1.0 ka BP) exhibit  $\Sigma$ IIIa / $\Sigma$ IIa ratios > 0.59, which is higher than the proposed limit for soil production (Xiao et al., 2016). Modern (

the lowest CA:LA ratio in this study, input solely *via* soil organic matter is unlikely. Albeit these sites likely exhibited a much larger hydrological catchment in the Holocene. Nonetheless this highlights the limitations of ΣΙΙΙα /ΣΙΙα in varved (and typical) lake systems (O'Beirne et al., 2024).

An alternative approach involves comparing the proportion of tetra-, penta-, and hexamethylated GDGTs alongside those from different environments, including soils, peats, and lakes (Dearing Crampton-Flood et al., 2020; Martinez-Sosa et al., 2021; Raberg et al., 2021; Bauersachs et al., 2024). For each site, our data reveals greater overlap with global lakes (as opposed to soils/peats), suggesting that brGDGTs in this study are largely produced within the lacustrine environment (Fig. 3). The BIGMaC machine-learning algorithm, on the basis of the fractional abundance of both isoGDGTs and brGDGTs in comparison with a global dataset (Martinez-Sosa et al., 2023), also suggests that most samples from Diss Mere (100%), Nautajärvi (67%), and Meerfelder Maar (100%) have GDGTs defined as 'lake-type'. Only the modern samples and two late Holocene samples (at 1.0 and 2.0 ka BP) from Nautajärvi are classified as 'peat-type' (this is also reflected in Fig. 4). Taken together, this suggests that the predominant source of brGDGTs in these laminated sediments is lacustrine.

#### 4.3 Impact of lacustrine processes on brGDGT distributions

Seasonal stratification and hypolimnetic anoxia/hypoxia are considered key additional controls on brGDGT distributions in lacustrine environments (Weber et al., 2018). In many stratified lakes, brGDGT production has been identified below the thermocline (Buckles et al., 2014; Miller et al., 2018; Sinninghe Damsté et al., 2022) with enhanced brGDGT production in low oxygen conditions (Weber et al., 2018; van Bree et al., 2020; Baxter et al., 2024; Zander et al., 2024). Changes in sediment chemistry associated with individual laminations also reveals the influence of different seasonal processes in lake environments, for example calcite precipitation in summer months and increased titanium in autumn and winter months relating to increased catchment sediment flux (e.g., Marshall et al., 2012; Zolitschka et al., 2015). To explore whether and/or how the brGDGT distributions in lakes in this study are influenced by specific limnological processes (e.g., stratification, anoxia, in wash), we compare our data to physical lake properties and μ-XRF measurements taken from the same sediment profiles.

#### 4.3.1 Redox influence upon brGDGT distributions

The lakes in this study are all thermally stratified in summer months. However, in terms of mixing regime, Diss Mere and

Meerfelder Maar were meromictic prior to 2 ka BP, whilst Nautajärvi is dimictic in the early and late Holocene with episodic
shifts towards a more meromictic regime coinciding with episodes of reduced and strengthened overturning in the midHolocene (Table 1; Lincoln et al., 2025). These mixing regimes influence the distribution of oxygen through the water
column, with each lake demonstrating hypoxic/anoxic bottom waters contributing to varve formation and preservation. In
permanently stratified (meromictic) lakes, pentamethylated and hexamethylated brGDGTs increase in abundance with depth
and are associated with anoxic conditions (Weber et al., 2018; Yao et al., 2020; van Bree et al., 2020). As the lakes in this

https://doi.org/10.5194/egusphere-2025-5701 Preprint. Discussion started: 27 November 2025

© Author(s) 2025. CC BY 4.0 License.

study exhibit anoxic/hypoxic hypolimneitic conditions in the Holocene, as a result of their mixing regime (above; and Table 1) high fractional abundances of pentamethylated (brGDGT-IIa, IIa') and hexamethylated brGDGTs (brGDGT-IIIa, IIIa') suggests a potential redox influence on brGDGT distributions.

In addition, we identify relationships between brGDGTs and redox sensitive elements across each site location (i.e. sulphur (S) and iron (Fe). In lakes insoluble sulphides can form through conversion of sulphate by sulphate-reducing bacteria (Luo, 2018) under anoxic conditions, whilst precipitation of Fe-hydroxide is associated with oxic conditions (Davison, 1993). At Nautajärvi, we identify a significant positive relationship between S<sub>clr</sub> and brGDGT-Ia (r = 0.45, p = 0.012) and brGDGT-IIa (r = 0.54; p = 0.002), and a significant negative relationship between Fe<sub>clr</sub> and brGDGT-IIIa (r = -0.52; p = 0.003) (S5).

Given that we observe opposing relationships between these redox-sensitive elements (Fig. 4) alongside temperature-sensitive brGDGTs (i.e., lower relative brGDGT-IIIa with high Fe, derived from mineral precipitates linked to phases of relatively greater hypolimnetic oxidation during strengthened mixing phases; Lincoln et al. (2025)), we suggest that hypoxic conditions and shifts to relatively more oxygenated conditions may influence brGDGT production at Nautajärvi (e.g., Raberg et al., 2022; 2025).

In contrast, at Diss Mere, we observe the opposite relationship between  $S_{clr}$  and brGDGT-IIIa (r = -0.46; p = 0.003) (S3). This is also reflected in the redundancy analysis with  $S_{clr}$  being positively loaded with  $Fe_{clr}$  at Diss Mere (Fig. 4). There is also a significant correlation between ln(Fe/Ti), which at Diss Mere captures the anoxic conditions of bottom waters (Boyall et al., 2024), and both brGDGT-IIa (r = -0.4; p = 0.01) and brGDGT-IIa' (r = 0.46; p = 0.004) (S3). Increased ln(Fe/Ti) indicates increased bottom water anoxia (Boyall et al., 2024). As such, the negative relationship between brGDGT-IIa and ln(Fe/Ti) suggests that anoxic conditions in the lake, due to incomplete mixing in a meromictic setting, result in lower relative brGDGT-IIa abundance. Whilst these patterns are observable at Diss Mere and Nautajärvi, we only observe a significant relationship between  $S_{clr}$  and brGDGT-IIa' at Meerfelder Maar (r = 0.36; p = 0.045) in terms of redox sensitive elements (S7). Therefore, at Meerfelder Maar, like Diss Mere, anoxic conditions are considered less important than at Nautajärvi for brGDGTs, which is a pattern that has emerged across dimictic and meromictic lake settings globally (Raberg et al., 2025).

Therefore, for certain temperature-sensitive 5-methyl compounds site specific conditions and the position of the oxycline appear important for GDGT synthesis. Whilst we consider anoxia as a factor, the scatter in the correlations suggests that there are further controls on brGDGT production (e.g., temperature).

#### 4.3.2 brGDGT response to limnological characteristics / processes within varved lakes

brGDGT distributions and cyclisation of branched tetraethers (CBT')-derived pH reconstructions differ across each site (Figs. 5, 6, 7) are likely influenced by differences in water chemistry which are associated with varve formation. At Diss

Mere, calcite precipitation occurs predominantly within the summer leading to the formation of calcite laminae. This is a product of alkaline lake waters being supersaturated with CaCO<sub>3</sub> throughout the year, with a greater degree of precipitation occurring during the summer months. Although our sampling strategy encompasses multiple varve laminations (i.e., both organic and calcite sub-layers), brGDGT-derived pH values are relatively alkaline throughout the Holocene ( $\sim 8.5 - 9.5$ ). We do not observe a significant correlation between CBT' and Cacir or CBT' and ln(Ca/Ti) at Diss Mere (S4), likely due to this sampling approach. Nonetheless, our pH reconstructions are aligned to pH readings (~ 9) from epilimnetic waters (Boyall et al., 2023). At Nautajärvi the winter season is represented by clastic lamina reflective of winter precipitation and snowmelt run off whereas the summer layer is related to lake productivity and allochthonous carbon yield from the catchment via surface runoff (Lincoln et al., 2025). This is consistent with more acidic conditions within the lake, as suggested by less alkaline pH reconstructions which is a product of the influx of acidic snowmelt during spring (e.g., Korkonen et al., 2017). However, we also observe a slight reduction in alkalinity from the early to late Holocene at Nautajärvi, which aligns with observations from other Scandinavian locations (Rosén and Hammarlund, 2007), and may be attributed to a gradual increase in input of acidic soil or peat, as is consistent with a switch to the machine-learning-defined 'peat' source for late Holocene brGDGTs. We also observe positive relationships between ln(Ca/Ti) (r = 0.55; p = 0.001), ln(Ca/K) (r = 0.54; p = 0.002) and CBT'/pH at Meerfelder Maar (S8), suggesting that variability in Ti<sub>clr</sub> and K<sub>clr</sub> (which mirror each other) impacts the 6-methyl isomerisation response (e.g., Peaple et al., 2022). Authigenic calcite only precipitates during a short period of the early Holocene in Meerfelder Maar (Martin-Puertas et al., 2017). The strong Cacir RDA loadings (Fig. 4) are driven by early Holocene samples only and are not, therefore, considered a driver of the Holocene GDGT response.

Like CBT', the relative proportion of 6-methyl brGDGTs is variable between sites, with Diss Mere and Meerfelder Maar exhibiting a greater contribution than Nautajärvi. This is reflected in average isomer ratio (IR<sub>6ME</sub>) values from the three sites (0.60, 0.41 and 0.15 at Diss Mere, Meerfelder Maar, and Nautajärvi, respectively). We observe a significant relationship between IR<sub>6ME</sub> and ln(Ca/Ti) at both Diss Mere and Meerfelder Maar (S4, S8), which at the latter site appears predominantly driven by the samples from within the early Holocene. This suggests that brGDGT isomerisation responds to annual CaCO<sub>3</sub> saturation within lake waters (Diss Mere) and changes to winter mineral flux (Meerfelder Maar) and therefore reflect site dependent autochthonous and allochthonous processes. Given the low IR<sub>6ME</sub> from Nautajärvi we do not discuss this further here.

We show that differences in the distribution of brGDGTs and metrics can be related to processes that operate in seasonally stratified lakes. The data suggests that anoxia, limnological and catchment processes can influence brGDGTs in these environments but that different processes impact brGDGTs at each site. However, given the imperfect correlations other factors (e.g., climate/temperature) remain seen as a key component (Section 4.4).

#### 4.4 Reliability of brGDGT-derived temperature reconstructions from varved lake sediments

#### 4.4.1 Modern brGDGT reconstructions consistent with instrumental observations

To ascertain whether brGDGT temperature reconstructions can be used to infer Holocene climate variability, we compare modern (<300 year) brGDGT-based temperature reconstructions to instrumental data from Diss Mere and Nautajärvi (Fig. 8). This same analysis is not possible from Meerfelder Maar owing to disrupted core top sediments. brGDGT-based mean temperature of months above freezing (MAF) reconstructions from Diss Mere are similar to long-term instrumental mean annual air temperature (MAAT) data, but are ~1°C to 2°C warmer than instrumental data between 1882 and 1950 AD but converge more closely after 1950 AD. As Diss Mere is not seasonally frozen, brGDGT-derived MAF estimates approximate MAAT albeit, on average, appear slightly warmer in modern settings.

At Nautajärvi, brGDGT temperature reconstructions overestimate instrumental MAAT by > 5°C when applying a global lacustrine MAF temperature calibration. The BIGMaC machine-learning algorithm suggests that brGDGTs from samples covering the last 300 years at this site could be derived from 'peat' rather than within the lake itself (see Section 4.2) and that a lacustrine calibration in these modern samples may be inappropriate. To explore this further, we recalculate temperatures at Nautajärvi using a combined global peat and soil dataset (Dearing Crampton-Flood et al., 2020) calibrated to either MAAT or MAF. However, MAAT reconstructions yield temperatures that are consistently lower (3°C - 5°C) than instrumental MAAT. Instead, we resampled instrumental MAAT to MAF to only include months where the lake is sub-aerially exposed and find a stronger overlap between the brGDGT MAF temperatures and instrumental data. This suggests that brGDGTs at Nautajärvi are recording warm season temperatures and that a lake calibration is appropriate. This shows that modern lake calibrations approximate modern instrumental observations from these two locations, but reconstructions need to take into consideration MAAT or MAF calibrations depending on location.

#### 4.4.2 Effect of 6-methyl brGDGT isomers on Holocene MBT'5ME values

Previous work has suggested that application of the methylation index of branched tetraethers (MBT' $_{5ME}$ ) in lakes with abundant 6-methyl brGDGTs can yield lower- or higher-than-expected temperature estimates (Dang et al., 2018; Russell et al., 2018; Bauersachs et al., 2024). Although the causes of this discrepancy remain unclear, it is considered to reflect a non-thermal response (Novak et al., 2025). There is no significant correlation between IR $_{6ME}$  and MBT' $_{5ME}$  at either Meerfelder Maar or Nautajärvi, respectively (r = 0.01; p = 0.97, r = -0.36; p = 0.053). Further, only samples from Diss Mere exceed the proposed IR $_{6ME}$  threshold of 0.5 where temperature estimates may be unreliable (Bauersachs et al., 2024). At Diss Mere, there is a significant correlation between IR $_{6ME}$  and MBT' $_{5ME}$  (r = 0.65; p = 

Figure 8: Modern temperature reconstructions from Diss Mere (left) and Nautajärvi (right). Shown are left) GDGT-based MAF reconstructions at Diss Mere (black; Martinez-Sosa et al. (2021), green; Raberg et al. (2021)); and reconstructed GDGT temperatures versus a) modern MAAT instrumental data from both the Lowestoft meteorological station (light blue) and from 5 km gridded temperature data using the HadUK dataset (light orange; Hollis et al., 2019) and right) GDGT-based MAAT soil reconstruction (orange; Dearing Crampton-Flood et al. (2020)), a GDGT-based MAF soil reconstruction (yellow; Dearing Crampton Flood et al. (2020)) and GDGT-based MAF lake reconstructions (black; Martinez-Sosa et al. (2021), green; Raberg et al. (2021)). These data are shown against modern MAAT instrumental data from both the Heinola Asemantaus (light blue) and Helsinki Kaisaniemi (light orange) meteorological stations which have been re-sampled in the far-right panel to show only months where the lake is sub-aerially exposed (May to November).

this data incorporates sparsely sampled datapoints that track low-frequency climate variability (see Section 4.5). If the data is subset to remove the influence of low-frequency variability and includes only highly sampled mid-Holocene data, the IR<sub>6ME</sub> and MBT'<sub>5ME</sub> relationship is much weaker (r = 0.27; p = 0.141).

The observed similarities with regional reconstructions (Section 4.5) and instrumental data (Section 4.4.1), suggest that whilst there may be a minor influence, it is unlikely that the brGDGT distribution at Diss Mere is controlled by non-thermal factors associated with  $IR_{6ME}$ . Our results thus emphasize that the 0.5 threshold may vary on a site-by-site basis. Indeed, a lower cutoff of 0.4 was recently proposed for Lake Baikal (Novak et al., 2025).

# 4.5 New insights into terrestrial temperature evolution during the Holocene

Above we discussed the probable source of different GDGTs in annually laminated lake archives (Section 4.1 - 4.2), the influence of different lake processes on brGDGTs (Section 4.3) and the overall reliability of brGDGT reconstructions in modern and Holocene contexts (Section 4.4). Below we show the comparison of brGDGT reconstructions from our lakes to regional Holocene climate observations (Fig. 9). As reconstructions of MAF at Diss Mere (11-15°C) and Meerfelder Maar (11-14°C) are both similar and warmer than at Nautajärvi (8-10°C) (Fig. 9), our brGDGT temperature reconstructions are, in the first order, performing as expected in light of site latitudinal differences. However, differences can be observed regionally.

#### 540 4.5.1 Temperature evolution across the Holocene

Diss Mere and Meerfelder Maar exhibit  $\sim 2^{\circ}\text{C}$  -  $3^{\circ}\text{C}$  warming between 10 and 8 / 7 ka BP, respectively. Peak warmth occurs at Diss Mere at  $\sim 8$  ka BP whilst this occurs between 5.6 and 4.3 ka BP at Meerfelder Maar (with a similar temperature at  $\sim 3$  ka BP; albeit for a single datapoint in a sparsely sampled section). At Diss Mere temperatures decrease from 8 ka BP (depending on calibration), with lowest temperature recorded in the late Holocene at  $\sim 2$  ka BP. Low temperatures are also recorded at Meerfelder Maar at 2 ka BP. In contrast, Nautajärvi reveals only a small temperature increase from the early to mid-Holocene from 9.8 ka BP, mid-Holocene peak warmth between 6 and 4.8 ka BP and rising temperatures across the late-Holocene (Fig. 9).

The general increase in temperature from the early Holocene and decline in the late Holocene as observed as Diss Mere and partly at Meerfelder Maar is comparable to global temperature trends across the Holocene (e.g., Fig. 9; Kaufman et al., 2020; Salonen et al., 2024) and follows patterns of precessionally driven summer insolation (Loutre et al., 2004). However, increasing temperatures at Nautajärvi in the late Holocene appears at odds with both this pattern and Arctic neoglacial cooling (e.g., McKay et al., 2018). Orme et al. (2018) identify warming sea surface temperatures from 2 ka BP from the Greenland and Norwegian Seas and the Iceland Basin, which adds to evidence of warming from the south east Norwegian Sea, warm climate anomalies in Scandinavia and a greater abundance of warm diatom taxa in the Baltic Sea (Seppä et al., 2009; Sejrup et al., 2016; van Wirdum et al., 2019). Despite this, it is challenging to reconcile opposing patterns in Europe, especially given that Nautajärvi is trending towards warmer climates than in the mid-Holocene. Therefore, at Nautajärvi ecological changes to bacterial communities, via changes in GDGT source in the mid – late Holocene, and changes in recorded seasonality of temperature cannot be discounted.

Despite these proxy observations, differences in the long-term evolution of temperature (e.g., in the timing of peak warmth) between regions reveal a spatial signal with sites in mainland Europe revealing differences compared to the UK. We suggest that the more maritime location of Diss Mere may be more sensitive to northern North Atlantic conditions, which,

Figure 9: Comparative Holocene climate records. a) summer insolation at 65°N; b) Holocene global surface temperature anomalies (Kaufman et al., 2020); c) a pollen-based multi-method ensemble of Holocene July temperatures (Salonen et al., 2024); d) brGDGT multiple linear regression MAF temperatures, northern Finland (Otiniano et al., 2024); e) brGDGT multiple linear regression MAAT temperatures, Massif-Central, France (Martin et al., 2020); f) brGDGT Bayesian MAF temperatures, eastern Carpathians, Romania (Ramos-Román et al., 2022); g) brGDGT Bayesian MAAT temperatures, Massif-Central, France (d'Olivera et al., 2023); and h, I, j) the Bayesian lake MAF (black; Martinez-Sosa et al., 2021) and methylation set multivariate 'regression MAF reconstructions (green; Raberg et al., 2021) from sites in this study.

with sea surface temperature records, depending on location and proxy, also reveal earlier peak warmth (e.g., Andersson et al., 2010; Berner et al., 2011; Cartapanis et al., 2022). However, why peak warmth is delayed at Meerfelder Maar, is not yet clear. At Nautajärvi mid-Holocene peak warmth is not surprising given similarities to a multi-ensemble pollen reconstruction (Salonen et al., 2024) and a brGDGT reconstruction (Otiniano et al., 2024) from sites in northern Finland (Fig. 9). In Finland delayed peak warmth may be related to the influence of the final wastage of the Fennoscandian ice sheet (Patton et al., 2017) and associated disruption of ocean circulation patterns (Cartapanis et al., 2022). Globally, at latitudes of >60°N, continental sites display peak warmth either prior to 10 ka BP or between 8 and 4 ka BP (Cartapanis et al., 2022). At 10 ka BP, Nautajärvi was a component of Lake Ancylus and connected to the Baltic Sea basin (Ojala et al., 2005) and therefore was not a competent lacustrine archive at this time. Therefore, regional temporal differences in temperature evolution appears variable between more maritime locations and mainland Europe.

This spatial pattern is also observed over the mid-Holocene. Previous west-European GDGT-based reconstructions reveal a general increase in temperatures across the mid-Holocene (e.g., van den Bos et al., 2018; d'Oliveira et al., 2023; Fig. 9). Whilst reconstructions from a site in eastern Europe demonstrates cooling trends (e.g., Ramos-Román et al., 2022). Further reconstruction complexity is added with a western European site displaying warming in the mid-Holocene (Martin et al., 2020). However, this has been suggested to relate to differences in reconstructed temperature seasonality alongside localised site differences, for example in environmental contexts and altitude (d'Olivera et al., 2023). Nonetheless, the trends in the temperature reconstructions from both Diss Mere and Meerfelder Maar are comparable to European Holocene brGDGT-reconstructions (Fig. 9), albeit with local and/or spatial influences (e.g., site elevation, continental position) seemingly being important in the direction of temperature change during the mid-Holocene.

#### 4.5.2 mid-Holocene variance

By exploiting varved lake sediments, we show that during the mid-Holocene, Diss Mere and Meerfelder Maar median reconstructions exhibit ~ 2°C - 3°C of temperature variability at multi-decadal scales. Comparisons to both pollen and brGDGT-based reconstructions from the European mid-latitudes reveal a similar range of variability (e.g., Seppä et al., 2009; Martin et al., 2020); albeit with some records displaying increased variability (~4°C - 5°C) at multi-decadal (Ramos-Román et al., 2022) and centennial scales (d'Oliveira et al., 2023) (Fig. 9). At Nautajärvi, temperatures are stable during the mid-Holocene (Fig. 9; median variability of 1°C). This is comparable to the brGDGT reconstruction from northern Finland between 7 and 4 ka BP (Otiniano et al., 2024). However, this is incongruent with non-brGDGT based temperature observations from Scandinavia (e.g., Shala et al., 2017; Salonen et al., 2024) where greater temperature variability (2°C - 3°C) is observed. Our reconstructions reveal a mostly comparable range of variability to other brGDGT reconstructions and suggest that higher latitudinal locations exhibit less variance than lower latitude locations. However, as reconstruction amplitude can be different between GDGT and traditional proxy reconstructions (pollen and/or chironomid) it is likely that

https://doi.org/10.5194/egusphere-2025-5701 Preprint. Discussion started: 27 November 2025

© Author(s) 2025. CC BY 4.0 License.

EGUsphere Preprint repository

some of the controls discussed above may be important in these lake settings (Section 4.3), or the seasonality of temperature recorded varies by proxy and by latitudinal setting.

4.5 Conclusion

We show that isoGDGTs are strongly influenced by methanogenesis linked to hypolimnetic hypoxia and that TEX<sub>86</sub> derived LST reconstructions are unreliable in annually laminated lake systems (or lakes that exhibit hypoxia/anoxia). In contrast, we show that brGDGTs are likely synthesised *in-situ* within each lake. Whilst there appears to be a redox influence at each location, the degree to which oxygen depletion affects brGDGT-temperatures appears minimal. This is similar to limnological processes that influence lamination style and limnological characteristics (e.g., pH) in each lake.

We show that there is a good correspondence between brGDGT temperatures and modern instrumental data, demonstrating suitability for brGDGT temperature reconstruction at each location. Further, the reconstructed temperatures from Holoceneaged sediments at Diss Mere and Meerfelder Maar resemble regional climate reconstructions (partly at Meerfelder Maar) and exhibit similar multi-decadal temperature variability. Nautajärvi exhibits limited temperature variability - this is comparable to another brGDGT reconstruction from Finland. However, there is mismatch between brGDGT and regional reconstructions during the late Holocene at this site (3 ka BP to present), suggesting other confounding factors may influence brGDGTs during the late Holocene at Nautajärvi.

Taken together, we show that annually laminated lake sequences are valuable archives for brGDGT-based climate reconstruction. However, we also show that processes that are specific to these unique laminated archives can lead to a brGDGT response. We therefore suggest that GDGT-based reconstructions be compared with additional site data or regional climate observations to disentangle the influence of lake process. In doing so, GDGT-based reconstructions from annually laminated records have the potential to deconvolve both multi-decadal climate variability and if performed at higher temporal resolutions, seasonal climate dynamics.

Data availability

All data associated with this article is available at Zenodo: https://doi.org/10.5281/zenodo.17610784

Author contributions

AMA, GNI and PGL conceptualised the study. AMA, GNI, and PGL formalised the methodology. AMA performed the analysis with input from MRB, DF, IB and HW. AB, PL, AEKO and CMP were involved in data curation. AMA wrote the original manuscript with input in GNI and PGL. All authors contributed to the review and editing process.

#### **Conflict of interest**

The authors declare that there are no competing interests in this research.

#### Disclaimer

The authors declare that this work has not been published nor is under consideration anywhere else.

## Acknowledgements

This study was funded by the UKRI Medical Research Council through a Future Leaders Fellowship held by C.M.-P.:

DECADAL: Rethinking Palaeoclimatology for Society (MR/W009641/1) and further supported by a UKRI Natural Environment Research Council National Environmental Isotope Facility (NEIF) grant (NEIF\_2602.0423) to G.N.I, A.M.A and P.G.L. The authors wish to thank the UKRI Natural Environment Research Council (NERC) for partial funding of the National Environmental Isotope Facility (NEIF; contract no. NE/V003917/1). G.N.I. is supported by a GCRF Royal Society Dorothy Hodgkin Fellowship (DHF\R1\191178) with additional support via the Royal Society (RF\ERE\231019, RF\ERE\210068). The authors thank the Maarmuseum in Manderscheid for logistical support during coring of Meerfelder Maar, with work at Nautajärvi conducted with the collaboration of Digital Waters Flagship (DIWA) (decision no. 359247) funded by the Research Council of Finland. The authors would like to thank Dr. Saija Saarni and Emilia Kosonen, who assisted with core extraction at NautajärvI, Dr Sargent Bray and Dr Mark Peaple for discussions around analytical chemistry and GDGT-techniques and the DECADAL project team for discussions around the manuscript.

# 650 Financial support

This work was supported by the following funding sources:

UKRI Medical Research Council through a Future Leaders Fellowship (MR/W009641/1)

UKRI Natural Environment Research Council National Environmental Isotope Faciliy (NEIF) grant (NEIF 2602.0423)

UKRI Natural Environment Research Council partial funding of the National Environmental Isotope Facility (NEIF;

NE/V003917/1).

GCRF Royal Society Dorothy Hodgkin Fellowship (DHF\R1\191178)

The Royal Society (RF\ERE\231019, RF\ERE\210068)

Digital Waters Flagship (DIWA) (no. 359247)

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
