# Peer review of "Evaluating glycerol dialkyl glycerol tetraether (GDGT)-based reconstructions from varved lake sediments during the Holocene"

_EGUsphere, 2025_

## Author Comment (AC1)

Response to reviewers

Dear Sebastian Naeher and Ashley Abrook,

This manuscript presents measurements of iso- and br-GDGTs in three Holocene varved sediment sequences in northern Europe, as well as at higher resolution for the last 300 years for two of the sites, allowing comparison with instrumental records. An impressive amount of data is presented, and the manuscript is well written with good structure. By presenting three new Holocene records, this manuscript is a valuable contribution to paleoclimate knowledge in northern Europe, and the investigation of GDGTs in varved lake sediments is of clear relevance to the proxy community. However, some interpretations, particularly for sparsely sampled late Holocene intervals, are not robustly supported by the data. A deeper discussion of discrepancies between GDGT based reconstructions and instrumental records would also improve the manuscript. Provided that these points and the more detailed comments below are adequately addressed, I see no issues with publication of this manuscript.

We would like to thank the reviewer for their positive review and for noting the 'impressive amount of data' that is presented, and that our manuscript is of 'clear relevance to the proxy community'. We also thank the reviewer for their constructive comments, which will undoubtedly improve the manuscript. Below we provide answers to each of the comments after they appear in red.

**General comments:**

- Why are the last 300-year data not included in the Holocene reconstructions presented in figs 5, 6, 7 and 9? It would be very useful to extend the long records towards the present if possible. If suitable, I would suggest adding it, or clarify in the manuscript why it is not suitable.

  This is a good and very valid point. We left our the last 300-year data from the long-term view as these systems are different in modern settings compared to the Holocene. For example, at Diss Mere the system is not varved from 2ka BP and is presently eutrophic. At Nautajarvi whilst the lake is still accumulating varves in the present day, human impact in the Finnish region of the lake began roughly 2- 2.5 ka BP. So we thought it best to separate out the disturbed / undisturbed parts of the record. Additionally, as you have pointed out, there are some caveats with the Diss Mere record in relation to the instrumental data (Nautajarvi follows instrumental data well). Nonetheless we will add the last-300 year data into Figures 5, 6 and 7 to extend the Holocene record to the present.

- The high resolution recent (last 300 year) data that is compared to instrumental data is very interesting. Since one of the cores follow the instrumental data, and the other does not, I think this section deserves more attention and some further discussion would be useful, as mentioned below. Can the findings from the core-tops be further used to indicate which records are more robust down-core?

  We thank the reviewer for this comment and agree that this deserves more discussion. We demonstrated that brGDGT temperature reconstructions align closely with modern instrumental data at Nautajarvi, and whilst brGDGT reconstructions approximate instrumental data at Diss Mere, there are periods where this alignment is less strong (Section 4.4). We do want to note though that although the alignment is less strong the instrumental data are within calibration error of the reconstruction. At Diss Mere, we consider that the alignment may be less strong due to a number of factors, including sedimentation rate increases, a shift in diatom and blue-green algal communities and changes in catchment vegetation leading to eutrophication from 2 ka BP onwards, especially with intensified human activity over the last 1 kyr (see Boyall et al., 2023; 2024). These changes are driven by an increase in detrital input following continued human occupation (Boyall et al., 2023; 2024).

  We suggest this may have altered the microbial community and could yield slightly different brGDGT temperature estimates (as observed elsewhere; e.g. Russell et al., 2018). We will therefore expand this section to explore why there are different relationships between brGDGT-derived and instrumental temperature data at Nautajarvi and Diss Mere.

- Section 4.5.1 discussed the temperature trends over the Holocene based on the three new records. Conclusions are made based on the changes in trends over the later Holocene, but the records only contain 2-4 data points over this period, leading to very high weight being given to single datapoints. Given that there can be large scatter in GDGT data, this is not robust, and the section should be reworked. Potentially this can be improved by adding the last 300 year data as mentioned above.

  We thank the reviewer for this comment - we acknowledge the sparse nature of some parts of the record. However, we have published independent climate data from these locations (using the same sample) that provide support for our long-term Holocene interpretations. For example, the Holocene calcium curve from Diss Mere (interpreted as reflecting a summer signal in Boyall et al., 2024) evolves in a similar fashion to our brGDGT temperature reconstruction. Similarly, the trends in XRF data and palynological growing degree day reconstructions from Nautajarvi (Ojala et al., 2008; Lincoln et al., 2025) follows the brGDGT reconstructions over the early- and mid-Holocene. However, there is a clear

mismatch in the late-Holocene where the brGDGTs indicate warming and we suggest that higher resolution data is required to explore this further (this is currently being investigated outside of this manuscript. To support our interpretations of the GDGT data we will provide additional site reconstructions in the supplement.

- Is TOC data available from the studied cores? Comparing the GDGT indexes with TOC may be very useful, since organic contents may be an important variable that covaries with GDGT distribution changes. Sudden shifts in TOC contents are strong indicators of environmental and limnological shifts, and are therefore likely to influence also GDGT distributions, see for example Hällberg et al., 2023 (https://doi.org/10.1016/j.orggeochem.2023.104702).

  Unfortunately highly resolved TOC data is not available from these sediments. We only have spot loss-on-ignition data from Nautajarvi. Whilst TOC is absolutely an important additional variable to consider in this type of work with large swings in depositional environment (e.g., peat-to-lake, or soil-to-peat; e.g. see Inglis et al., 2019), our lake sediments are continuously varved across most of the Holocene (until 2ka at Diss Mere). As the style of deposition is rhythmic and stable throughout the Holocene, this implies largely stable TOC values. Whilst the thicknesses of individual varves do change (e.g., Martin-Puertas et al., 2025), this is unlikely to be driven by large shifts in TOC and is more reflective of more features inherent to the climatic season (length/persistence; see Martin-Puertas et al., 2025). Where varves are not preserved (last 2 ka at Diss Mere) TOC is likely to increase, however only two samples in our data cover this period.

- 7 methyl and GMGT data from these sites would be interesting to see from these sites, if that data is available. It may provide clues to the provenance of the GDGTs as well as potentially providing an additional indication of temperature variability (Baxter et al., 2019, https://doi.org/10.1016/j.gca.2019.05.039). Of course, this manuscript is already long, and 7-methyl brGDGTs and GMGTs should only be included if the authors deem this to have large explanatory power.

  We agree that 7-methyl GDGTs and GMGTs can provide additional insights into the bacterial community, but this is beyond the scope of this manuscript. The primary aim here is to test whether brGDGTs are robust temperature proxies in varved lakes. However, we acknowledge that GMGTs could be evaluated in future studies from these locations. 7-methyl GDGTs are likely to indicate changes in salinity which given the temperate locations of these lakes we do not consider as major contributors.

**Specific comments:**

- Fig 1:

  - in the B inset, lake depth isobars are not numbered as in a) and c) insets. This would be good to add.

    We will add this into these figures.

  - The brown symbols in depositional model for a) is not indicated in the legend what they are.

    Together with the green brush stroke these are organic remains. We will add this into the legend.

  - It is also not clear in the figure text why Diss Mere has three panels for the depositional model, so please add that explanation there. What does the light brown/green shading indicate?

    Diss mere has three panels as these are the differences in sediment facies for each of the lamination types. There are three varve types at Diss Mere represented by differences in sub-layer. Light brown / green also represents differences in sub-layer (calcite vs organic). The varve models for Nautajarvi and Meerfelder Maar, are, in contrast, very simple and stable. We will add this explanation into the figure text.

  - Letters in the lake insets for a) and b) are not explained in the figure text, and what are yellow markers in b)? If they are important in the manuscript, please explain them in the figure text. If not important, perhaps remove from figure?

    The letters relate to the different coring locations which we will add into the text. The yellow markers are other survey points and are not too important for this manuscript. We will remove them.

- Site description sections. Please provide references and explanation for the meromictic conditions at the lakes already here.

  We will add descriptors to the table that demonstrate mixing regime for each site and provide a reference for each from the lakes.

  Meromictic – No lake water overturn periods, constant lake stratification (Martin-Puertas et al., 2021)

  Dimictic - Two lake water overturn periods annually (Lincoln et al., 2025)

- How does the 0.5 cm resolution compare with the varve thickness? It would be interesting to know if it is technically possible to reach sub-annual resolution, and if some samples represent that in this study.

  The 0.5 cm resolution contains a number of varves which change throughout the record and between site locations. All of the lake sites have sub millimetre scale varves throughout with an approximate minimum number of varve years of 5 per 0.5 cm. We do not have any varve layers which approach 0.5 in thickness. This type of analysis would be interesting at those sites where thicker varves are preserved.

- The manuscript refers to GDGT-0 vs crenarchaeol as %GDGT-0. I would expect that to refer to the fractional abundance of GDGT-0 (relative to all isoGDGTs). Instead, why not refer to it simply as GDGT-0/cren, as frequently done previously, for instance by Baxter et al., 2021 https://doi.org/10.1016/j.quascirev.2021.107263 ? This would make it much clearer what is meant.

  Thank you for this. We use %GDGT-0 as originally defined in Sinninghe Damsté et al. (2012) as this enables simpler inter site comparisons.

  $$\% \, GDGT - 0 = (GDGT - 0 \, / \, (GDGT - 0 + Crenarchaeol)) \cdot 100$$

  The Baxter et al. (2021) approach is similar but the ratio is theoretically 'limitless' which makes comparisons between locations difficult. For the GDGT-0/cren ratio a value >2 is indicative of methanogen contribution. For %GDGT-0, a value > 67% is indicative of a methanogen contribution. Regardless of the metric we use, the key conclusions are identical (i.e. each lake is dominated by methanogenic archaea). However, we will add a statement to this formula stating that this should not be confused with the fractional abundance of GDGT-0.

- Refer to Hopmans et al., 2004 https://doi.org/10.1016/j.epsl.2004.05.012. Since that original publication is used in marine settings only, I would suggest also referring to a study showing that it can also be used as an aquatic/land signal for terrestrial sites.

  We will add Hopmans et al., 2004 in as a reference for BIT. We will also add Baxter et al. (2021) to show that it can be used in terrestrial settings.

- Is this 'wrong' classification correlated with an increased TOC from that core? Since TOC has previously been found to strongly correlate with GDGT distributions elsewhere (see for example fig 3 in Hällberg et al., 2023), I think it would be valuable to show this data, as mentioned earlier, if available.

  Unpublished loss-on-ignition spot samples from Nautajarvi suggest there are no rapid changes in TOC during the Holocene. Crucially, the varve structures also

remain constant suggesting no significant changes in TOC content across the Holocene. So whilst TOC is important we do not consider it a driver here. We will add this statement to the manuscript in this section.

- *"Across all three sites the BIT index is very high and is > 0.92"*. A BIT value of 0.92 is a significant change from >0.99 which is normally found in soils/terrestrial sites, and I would therefore suggest that a value of 0.92 likely represents a significantly different GDGT distribution, likely resulting from some environmental change. However, looking at figures 5-7, I do not see any such low values, so perhaps this is a typo?

  Thank you for this observation. All of our 'Holocene' samples (i.e. those from 1ka to 10ka) have high BIT (Diss Mere = >0.98; Nautajarvi = >0.99 (the first sample in the record Is 0.96); Meerfelder Maar = >0.99). It is when we include the 'modern' samples from Diss Mere where this statement comes from. The BIT in the modern samples range from 0.92 to 0.98. We will add that distinction into the manuscript.

- Fig 4. What are the "modern" samples in a) and b)? Surface samples?

  The modern samples represent the samples covering the last 300 years. We will clarify that in the figure text.

- Fig 4. What are the plot labels in the RDA plots? Ages? In that case, I would suggest adding a fill color gradient based on that value, to increase readability.

  Yes the plot labels are the sample ages. We can add the colour fill of the individual points to the sample ages to show alignment with the data.

- Fig 5e. The arrow with "marine provenance" needs to be relabeled, since i assume you mean aquatic rather than marine?

  Thank you for this observation. We will change to aquatic for all figures.

- Is the value 0.9 correct? It looks like it's higher based on the figures.

  We assume this is in reference to the BIT. This value includes the modern samples as suggested in the comment above. We will change this to reflect Holocene samples only. 'Across all lakes, Holocene samples reveal a high BIT index (>0.96).'

- 375 section, on source attribution. Additional evidence for soil input can be derived from degree of cyclization, IR6-methyl and 7 methyl GDGTs as done by Martin et al., 2019. GMGTs and their isomers may also provide further clues and indicate bacterial community shifts, see Hällberg et al., 2023.

Many thanks for these comments on source attribution. We agree that 7-methyl GDGTs and GMGTs can provide additional insights into the bacterial community, but this is beyond the scope of this manuscript and were not routinely identified. We do observe GMGTs in some samples but this will form the basis of a subsequent manuscript. Our primary aim here is to test whether brGDGTs are robust temperature proxies in varved lakes.

We do calculate IR6me values, however we do not have terrestrial soil values as comparators. As we already use three / four approaches to disentangle provenance, which are mostly in agreement with each other, we do not feel IR6me will add too much to these observations.

- Please clarify how it is reflected in fig 4: "*Nautajärvi are classified as 'peat-type' (this is also reflected in Fig. 4)*"

Thank you for this observation. What we meant by this was that the samples that have different GDGT distributions are grouped together separately from the rest of the Holocene. For ease we will remove 'this is also reflected in Fig. 4'.

- Reference needed.

We are unsure where in text this refers to.

- It is probably useful to also mention the results of Baxter et al., 2021 (https://doi.org/10.1016/j.quascirev.2021.107263) from meromictic Lake Challa here.

We have added the Baxter et al., 2024 reference to the following sentence: 'In permanently stratified (meromictic) lakes, pentamethylated and hexamethylated brGDGTs increase in abundance with depth and are associated with anoxic conditions (Weber et al., 2018; Yao et al., 2020; van Bree et al., 2020; Baxter et al., 2024).'

We aren't 100% sure where you would like this discussed. We added the Baxter et al. 2024 reference as this details brGDGTs and the 2021 reference details only the isoGDGTs. We mention Baxter et al. 2021 in Section 4.1.

- Section 4.3.2, and in particular lines 468-475. CBT' calculation includes 6-methyl brGDGTs, and is therefore a mix of cyclization and isomerization, despite the (perhaps misleading) name *cyclization of branched tetraethers*. It would therefore be better to compare degree of cyclization (Sinninghe Damsté et al., 2009 https://doi.org/10.1016/j.gca.2009.04.022) and IR6me instead of CBT' and IR6me, to disentangle these compound influences.

Thank you for the comment. In section 4.3.2 we employ CBT' and IR6me to evaluate how the GDGT and XRF data respond to changes in pH. Both metrics

have a global, linear relationship with soil pH and thus are deemed most suitable here. Although the degree of cyclisation offers an alternative approach to assess pH (see Table 1; Baxter et al., 2024), it also contains 5- and 6-methyl brGDGTs in its formulation and does not overcome the caveats raised by the reviewer.

We will alter the text in lines 462 – 475 to clarify more specifically when we discuss CBT' and IR6me XRF correlations.

- Fig 8. The figure text is quite messy.

  - The panels should be a, b, c, d, since it's four panels.

    We will alter the panels to show a, b, c, d as requested.

  - Currently, panel b) is not specified in the text, only a).

    Thank you for this observation. We will edit the figure caption to include each of the panels numerator.

  - I would propose labelling the panels for easier readability, such as "MAAT", "Tmay-Nov" or similar.

    Thank you for this observation, we will alter the panel labels to reflect the reconstructed and instrumental temperatures.

  - Better to use common era (CE) instead of BC/AD?

    We agree this is better to use and will change accordingly.

- It would be interesting with a deeper discussion of your results from the short core presented in fig 8. Nautajärvi has a very good agreement with the instrumental record, but Diss Mere shows the opposite trend. Please elaborate on this, and potential causes for it. The sudden offsets in temperature around 1940 in Diss Mere may be useful in investigating this. What happens in the GDGT distributions (or other data) to cause this offset? After the offset, the reconstructed temperature is lower than before, contrary to instrumental data which show warming. Any clues to why?

  We thank the reviewer and agree that this section requires some deeper thought and discussion. As outlined above this offset could be due to various mechanisms. Firstly, these sediments are not varved and as a result may host a different bacterial community compared to the laminated sediments of the Holocene- the upper 50-years of sediment display different GDGT distributions compared to much of the Holocene (e.g., less GDGT-1a). Secondly, around 1940, we observe increases in IR6me and changes in the ΣIIIa /ΣIIa ratio which may equally reveal a slightly different GDGT community. Thirdly, there is evidence that the lake is eutrophic at this time (owing to human influences). These additional

environmental factors likely explain why the reconstruction does not fit the instrumental data as well as at Nautajarvi.

Of note, this feature of core-top cooling has previously been identified in lakes from the northern hemisphere (e.g., Miller et al., 2018; Zhao et al., 2021). The Miller et al. (2018) record also shows a mid-20th century temperature peak before declining, with greater methylation observed after the peak, which is similar to our observations. These authors also suggest shifts in the bacterial community may overprint the GDGT – temperature relationship.

We will add this discussion into the manuscript.

- Fig 9.

  - I would propose to show all records with the same y axis spacing, so that it is possible to see which sites show very little change versus larger change.

    This is a very good point and will help with latter sections of the manuscript. We will change the y-axes so that they match.

  - The arrows indicating trends appear quite arbitrary and require some better explanation.

    The goal here was to assist in the visual representation of trends for what we discuss in text. We are happy to remove these arrows however if the reviewer feels they do not add to the discussion.

- Specify if you mean the *trend* here rather than amplitude.

  I believe this comment is in relation to Section 4.5.1. Here we are referring to trends so will add this detail in.

- 541: "*Peak warmth occurs at Diss Mere at ~ 8 ka BP*" this seems to be based on only a single datapoint, and is only true for the MBT based calibration, but at odds with the Raberg calibration. The way I read that graph (9h), the Diss Mere reconstruction shows slightly lower temperatures at 10-9 ka BP and after ~3 ka BP, but this is based on very few datapoints. The period 8-4 ka BP has very slightly elevated but variable temperatures. The temperature at ~5.9 ka BP is for example the coolest of the full record. I therefore find the discussion of the Diss Mere temperature evolution to be lacking in robustness, and higher sampling resolution would likely be needed to draw these conclusions. That said, like mentioned earlier, if the near surface data are added to the full reconstruction, the trends for the Late Holocene may be clearer.

  We thank the reviewer for this comment and agree that the sparseness of the sampling interval in the early and late Holocene does not necessarily always

help with observations of temperature trends. Whilst we agree with the reviewer that temperatures between 10 and 9 ka and after 4 ka are based on few data points and care does indeed need to be taken to avoid over interpretation we can provide evidence here from other proxy data from these sites (which we will include as supplementary figures) that dominant climate evolution broadly matches published climate data from these sites.

The ca. 3°C temperature rise from 10 – 8 ka and the 3°C temperature decline at the end of the Holocene is also comparable in evolution to the observations from other GDGT and traditional proxy reconstructions in Europe and shown on the figure (e.g. Salonen et al., 2024; Otiniano et al., 2024; Martin et al., 2020), which we believe adds credence to our observations (combined with the additional site evidence that we can show).

We will however modify that whole paragraph to reflect the observations more precisely: 'From the available data, peak warmth occurs at Diss Mere between 8 – 4 ka BP with the warmest temperature at 8 ka BP observed from a single data point. Peak warmth at Meerfelder Maar appears between 6 – 3 ka BP with the observation at 3 ka BP again from a single data point. Whilst caution is required when interpreting sparsely sampled data points, our data is broadly comparable to climate proxy data obtained from these records (Supplementary Information) and the comparative Europe wide reconstructions from Figure 9.'

- The statements about peak temperatures around 5.6-4.3ka BP at Meerfelder Maar also doesn't appear robust, with at least the datapoint at around 3k showing comparable temperature, with only one datapoint after that showing a slight cooling.

  We will alter this text to reflect that data more precisely. As above paragraph.

- Specify what is meant by "this" at the start of the paragraph.

  We will remove 'This' and reform the first sentence to 'Spatial differences in reconstructed temperatures also appear in the mid-Holocene.'

- But it is highly uncertain how much of this 2-3 degree temperature variability stems from methodological uncertainty/scatter versus actual climate shifts. This needs to be mentioned in the text.

  Thank you for this comment, we will add this note to our manuscript. 'By exploiting varved lake sediments, we show that during the mid-Holocene, Diss Mere and Meerfelder Maar median reconstructions exhibit ~ 2°C - 3°C of temperature variability at multi-decadal scales. Whilst this 2-3°C temperature variability may be a product of calibration uncertainty, comparative climate

reconstructions reveal similar patterns in climate variance. Pollen and brGDGT-based reconstructions…'

**Technical or minor corrections:**

- The original reference for tex86 is Schouten et al., 2002
  https://doi.org/10.1016/S0012-821X(02)00979-2

  We will add this in.

- This sentence comes a little out of the blue and requires a little more clarification. Is this threshold applied in this study?

  We are sorry but we do not know which sentence is being referred to here.

- Equation 9. Typo. As written now, it mathematically makes no sense. It should be GDGT0/(GDGT0+cren).

  Thank you for this observation, we will correct this to the correct formula.

- Why capitalization of crenarchaeol?

  This is an oversight on our behalf and we will correct all mentions of crenarchaeol in text.

- Section 3.1. Please refer to figure numbers when presenting index results.

  We will add figure numbers into the manuscript when presenting results.

- Fig 4. To improve readability of this figure, I would suggest removing "brGDGT" in front of each compound. It is unambiguous that Ia, IIa etc. are brGDGTs. This can also be done for figure 2 axis labels.

  Thank you for this advice. We will tidy up Figure 4 labels by removing brGDGT from the PC. However we would like to keep the labels in Figure 2 as they are as we shift from iso to brGDGTs in each sub-figure.

- Fig s1. Add in text that it is based on TEX86.

  We will add into text that the reconstruction is based on TEX86

- Specify that you mean in *reconstructed* LST?

  We will add clarification that this is reconstructed LST.

- Figures s4, s6, s8. Please make the order of the elements the same in all figures. Currently, the s6 figure has other order than s4 and s8, which reduces readability.

We will change the order of the elements in S6 to match the other figures.

- "vs" doesn't need to be in italics. But if you decide to still do that, be consistent throughout. For example, it is not in italics on line 362.

  We will change all of the vs to be un-italicised.

- Reference should be in parentheses.

  We will change this.

- Replace imperfect with moderate, or similar.

  We will change this wording.